# Robust Recovery of Adversarial Examples

Tejas Bana[1]    Jatan Loya[2]    Siddhant Kulkarni[3]

## Abstract

Adversarial examples are semantically associated with one class, but modern deep learning architectures fail to see the semantics and associate them to another class. As a result, these examples pose a profound risk to almost every deep learning model. Our proposed architecture can recover such examples effectively with more than $4\times$ the magnitude of attacks than the capability of the state-of-the-art model, despite having lesser parameters than the VGG-13 model. It is composed of a U-Net with the characteristics of self-attention & cross-attention, which enhances the semantics of the image. Our work also encompasses the differences in the results between Noise and Image reconstruction methodologies.

## 1. Introduction

Adversarial examples are machine learning and deep learning model inputs that an attacker has purposefully designed to cause the model to make a mistake. They are similar to optical illusions for machines. A human is clearly able to understand the semantic meaning conveyed by an image-based adversarial example. Neural networks tend to create exponential number of classification boundaries, which are very sensitive to perturbation. These examples defeat the whole purpose of deep learning since even a tiny perturbation in an image can lead to misclassification rendering the entire infrastructure useless.

These misclassifications could lead to loss of life in safety-critical applications like self-driving cars, where attackers may use stickers or paint to construct an adversarial stop sign. The vehicle would misinterpret as a 'yield' or other

signals. Attackers can use targeted attacks to make the model predict their desired output. Adversarial examples are tough to fight against since creating a theoretical model of the adversarial example generation process is challenging. Some of the most prevalent methods to generate adversarial examples are Fast Gradient Sign Method (FGSM) (Goodfellow et al., 2014), and Random with FGSM (R-FGSM) (Tramèr et al., 2017).

Our work aims at tackling the problem of recovering adversarial examples by generating negative noise and reconstructing the entire image. Previous work for recovering adversarial examples (Liao et al., 2018) only recovers a limited range of attacks which excludes heavily attacked examples (max perturbation in the current state-of-the-art method is 16). Some effective techniques (Liao et al., 2018) are known to be computationally expensive. Our proposed architecture can reverse a large range of adversarial attacks, including heavily attacked examples (Max perturbation of 64, $4\times$ more than ever achieved), which have lost their semantic meaning. Our proposed model can be easily added to an existing application pipeline without changing the existing models.

Our contributions through this paper include:

- A novel GAN architecture and training methodology to recover FGSM and R-FGSM adversarial examples irrespective of the magnitude of the attack, performing better than previous state-of-the-art on the CIFAR-10 dataset (Krizhevsky et al., 2014).

- A novel training procedure with adversarial training followed by post-training, which uses hybrid losses.

- The proposed generator has fewer parameters than a basic classification model like VGG-13 (Simonyan & Zisserman, 2014).

- Improving the semantics of the image after recovery and attaining recovered-classification accuracy of 98.5% as compared to 93% for original CIFAR-10 examples.

[1]Department of Information Technology,D.Y. Patil College of Engineering, Akurdi, India [2]Department of Computer Engineering, Vishwakarma Institute of Technology, Pune, India [3]Department of Computer Science, BITS Pilani, Hyderabad, India. Correspondence to: Tejas Bana <tejasbana@gmail.com>, Jatan Loya <jatan.loya18@vit.edu>, Siddhant Kulkarni <kulkarni.siddhant36@gmail.com>.

*Accepted by the ICML 2021 workshop on A Blessing in Disguise: The Prospects and Perils of Adversarial Machine Learning.* Copyright 2021 by the author(s).

## 2. Background

Adversarial examples are generally produced by adding some calculated noise that is unrecognizable by humans but is enough to fool a classifier. Various types of adversarial attacks are: white-box attack where it is assumed that the attacker has complete information and access to the model, grey-box attack where the attacker knows the model parameters of the target model but does not know that the target model has an active defensive mechanism, and black-box attack where the attacker only knows the input and output of the model and does not have any other information.

**Fast Gradient Sign Method:** (FGSM) attack is a type of white-box attack where the attack uses the gradient of the loss with respect to the input data, then adjusts the input data to maximize the loss.

$$adv_x = x + \epsilon \times sign(\nabla_x J(\theta, x, y)) \qquad (1)$$

In the above formula, $adv_x$ represents an adversarial image, 'x' is input image (Original), 'y' input label (Original), $\epsilon$ is the magnitude of attack, $\theta$ is the parameter of the model, and 'J' is the loss function. $\nabla_x$J($\theta$,x,y) is the gradient of the loss with respect to 'x'.$\epsilon$ is the metric of the magnitude of the adversarial attack.

**Random - Fast Gradient Sign Method:** is variant of the Fast Gradient Sign Method attack is a more efficient attack compared to the iterative method since this attack requires a single gradient computation. This attack applies a small random perturbation before relinearizing the models' loss to escape the non-smooth vicinity of the data point. R-FGSM outperforms FGSM for the same perturbation norm.

$$x^{adv} = x' + (\epsilon - \alpha) \times sign(\nabla_{x'} J(x', y_t rue)) \qquad (2)$$

where

$$x' = x + \alpha \times sign(N(0^d, I^d)) \qquad (3)$$

## 3. Motivation

The current state-of-the-art (Liao et al., 2018) model uses U-Net architecture and has an extremely large number of parameters. It was trained to reverse adversarial ImageNet (Deng et al., 2009) images created using three different classification models and two attacks, FGSM and IFGSM, with an $\epsilon$ range of 1-16. Training to reverse attacked ImageNet $300 \times 300px$ size images requires immense computational resources. Since we did not have the required computational budget we focused on making the GAN (Goodfellow et al., 2020) model much more efficient in terms of the number of parameters and reversing a much larger $\epsilon$ distribution on the CIFAR-10 dataset. The proposed GAN model is trained to reverse CIFAR-10 adversarial images with an $\epsilon$ range of

1-128, attacked from 3 different models with FGSM and R-FGSM attacks.

## 4. Dataset

Adversarial images are needed for both training and testing of the proposed method. For the training dataset, we used 50k images from the CIFAR-10 training set ( 5k images per class). Multiple adversarial attacking methods were used to distort these images and form a training set of adversarial images. Different attacking methods, including FGSM and RFGSM, are applied to the following models: Pre-trained Inception-v3 (Szegedy et al., 2015), Resnet-18 (He et al., 2015), VGG-13 individually. For each training example, the perturbation level E is sampled uniformly from integers in $[1, E\_max]$, here $E\_max$ is the maximum possible value of $\epsilon$ for that experiment.

Dataset of adversarial examples is generated at the time of GAN training. Every step of the training, a batch of 96 images is split into 32 for each of the three attacking models. These 32 images are divided into 16-16 for two attacks FGSM and R-FGSM, for each model. Furthermore, every image in these 16 images is attacked with a different $\epsilon$. Finally, every image in a batch is attacked by a different model by different method at different epsilon, making the training stable for the task.

Producing adversarial examples while training step gives the benefit of not storing a large amount of adversarial data and, second, making each training image in a batch more random. The permutation of different adversarial examples that could be generated is as follows:

$$50,000 \; Images \times 3 \; Models \times 2 \; Attacks \times E\_max$$

This permutation represents the number of different training inputs that can be produced which equates to 38.4 million training images

To prepare the validation set, We used 10k images from the CIFAR-10 validation set (1k images per class). Then applied the same method as described above.

For final testing, $E\_max$ test sets are constructed for every $\epsilon$ in the range between 1 to $E\_max$. They are obtained from the same clean 10K images from the CIFAR-10 validation set but using only the FGSM attack.

## 5. Architecture and Training Procedure

We used Attention U-Net (Oktay et al., 2018) as a base generator architecture because it automatically learns to focus on image structures of varying shapes and sizes and can be easily integrated into standard CNN architectures such as the U-Net model with minimal computational overhead while increasing the model prediction accuracy.

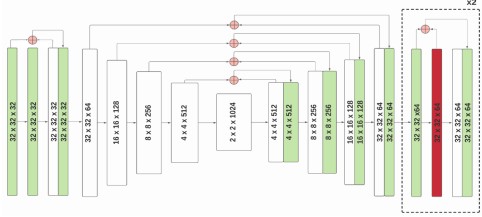

*Figure 1.* Generator Architecture (Red represents Self Attention Layer & Pink represents Cross Attention)

Added PixelShuffle rather than ConvTranspose, as it is an operation used in super-resolution models to implement efficient sub-pixel convolution with a stride of 1/r. Specifically, it rearranges elements in a tensor shape $(*, C \text{ x } r^2, H, W)$ to a tensor shape $(*, C, H \times r, W \times r)$. Outputs of the PixelShuffle layer are normalized using Batch Normalization and then passed through PReLU activation. Parametric Rectified Linear Unit (PReLU) is proposed to generalize the traditional rectified unit (ReLU). LeakyReLU was introduced to improve upon ReLU. It multiplies the negative input by a small value (like 0.01) and keeps the positive input as it is. PReLU allows the network to learn that small value during training so that our activation function can adapt to the other parameters (like weights and biases). The network can learn the slope parameter using backpropagation at a negligible increase in the cost of training.

Generally, models are made deep with more hidden layers to increase their prediction accuracy, essentially giving more parameters for the model to learn. Usually, 3-4 layers of convolutional layers are used at every stage of U-Net architecture. We instead used a single convolutional layer with a kernel size of 3 but used self-attention (Yang et al., 2019) at the decoding layers. The output of the Attention U-Net is passed through a self-attention layer followed by a residual cross attention connection.

### 5.1. Training Procedure

The model is trained using an adversarial procedure with three different loss functions: L1 loss (pixel-level loss), Adversarial loss from discriminator (MSELoss), and Classification loss from pre-trained CIFAR-10 Classifier (CrossEntropyLoss).

Even after adversarial training, there are tiny perturbations present in the generated image when compared to the original image; these perturbations may make the model yield the wrong prediction as these are progressively amplified by every passing layer of the network, known as the error amplification effect. The top layers of deep neural networks cause misclassification due to small perturbation. Training the model to minimize the difference of these top-level features can result in better predictions. These perturbations

can be removed by post-training methodology, where the generator trained by the adversarial procedure is guided by a loss function known as perceptual loss or feature matching loss.

Our experiments concluded that using two stages of training (adversarial training followed by post training) gives better results compared to including feature loss at stage one (adversarial training, which made the training unstable), GAN training was much more stable and converged faster.

## 6. Image vs Noise Reconstruction

Our work focuses on the differences in the results by the Image and Noise reconstruction procedure. GAN can be trained to reconstruct the original image with the input of an adversarial attacked image or can be trained to anticipate the negative noise, resulting in an original image when added to the adversarial image. Previous research works agree that noise reconstruction produces a better result when compared to image reconstruction, including the authors of the current state-of-the-art model for ImageNet image recovery. On the contrary, our results show that image reconstruction can still be the preferred choice depending upon the purpose of model implementation. Understanding the trade-off between these approaches is now more important than before because this is the fundamental choice while dealing with Diffusion Models (Ho et al., 2020), which has shown to work better than GAN. We have demonstrated the advantages and disadvantages of both approaches in Table 1

*Table 1.* Performance comparison for 3 methods

| Epsilon | Noise Reconstruction | | Image Reconstruction | | Image Reconstruction-2 | |
|---|---|---|---|---|---|---|
| | Inception-v3 | VGG-13 | Inception-v3 | VGG-13 | Inception-v3 | VGG-13 |
| 0 | 92.71% | 92.26% | 92.21% | 92.25% | 92.52% | 92.92% |
| 4 | 91.27% | 88.88% | 88.87% | 88.72% | 90.19% | 89.69% |
| 8 | 93.47% | 88.5% | 92.29% | 89.04% | 92.45% | 89.66% |
| 16 | 93.91% | 88.3% | 93.86% | 89.02% | 93.58% | 89.69% |
| 32 | 8.02% | 26.98% | 57.92% | 72.57% | 56.43% | 74.18% |

The result of column Image reconstruction-2 represents the scenario of image reconstruction with adversarial loss in post-training. The discriminator is not involved in post-training in the other two methods, hence only the feature loss as the training criterion.

From the results of Table 1, Noise reconstruction slightly gives higher accuracy but lacks robustness when dealing with an adversarial attack of higher $E\_max$, which lies outside the training distribution. We hypothesize that constructing noise works better than constructing images from perturbed images due to the bound on predictions. The range of the pixel values the generator has to predict while creating noise is much smaller [$-E\_max$, $E\_max$]. For image reconstruction, the generator has to encode all the relevant information and then reconstruct the image from the encoding. In this scenario, the generator has to predict the values

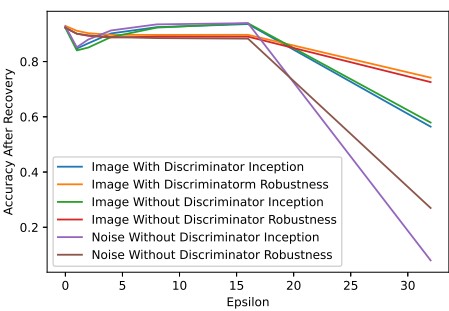

*Figure 2.* Performance Comparison of 3 methods

This proves that the generator can reverse the images and enhance the semantic meaning of the images, resulting in a performance boost in models. The results shows that the architecture can generalize and is robust.

*Table 2.* White Box Attacks

| Epsilon | Exposed VGG-13 | VGG-13 | Exposed Inception-v3 | Inception-v3 |
|---|---|---|---|---|
| 5 | 43.18% | 86.46% | 11.24% | 90.32% |
| 10 | 38.89% | 89.64% | 5.96% | 96.32% |
| 20 | 24.86% | 90.86% | 7.52% | 98.13% |
| 25 | 18.34% | 90.44% | 8.21% | 98.5% |
| 35 | 12.65% | 87.36% | 8.83% | 98.4% |
| 50 | 10.69% | 77.37% | 9.18% | 96.83% |
| 65 | 10.40% | 67.19% | 9.26% | 91.85% |

Black-Box attacks were performed with the adversaries as DenseNet-121 (Huang et al., 2018) and VGG-11; after the recovery, they were reclassified by Inception-v3. In Table 3, Exposed 1 and Recovered 1 represent the attacking model as DenseNet-121 and similarly for VGG-11. All the accuracy represents the classification accuracies of Inception-v3.

*Table 3.* Black Box Attacks

| Epsilon | Exposed 1 | Recovered 1 | Exposed 2 | Recovered 2 |
|---|---|---|---|---|
| 5 | 70.22% | 79.82% | 64.84% | 77.78% |
| 10 | 53.88% | 79.26% | 46.85% | 76.35% |
| 20 | 27.51% | 79.01% | 24.96% | 76.63% |
| 25 | 20.61% | 77.40% | 19.39% | 75.46% |
| 35 | 14.76% | 68.86% | 14.69% | 66.16% |
| 50 | 11.03% | 59.18% | 12.04% | 57.28% |
| 65 | 9.67% | 55.66% | 10.78% | 54.32% |

for each pixel in the range of 0 - 255 ([-1,1] since the use of Tanh), which is a larger scale than that of constructing noise, as perturbation added is only between E [-16,16], the pixel values of negative noise will also lie between this range.

This advantage of producing noise does not work well when trained to reverse data with higher perturbation. For example, $\epsilon = 128$, but at the same time, these perturbed examples cannot be termed as adversarial examples but classified as rubbish class examples (Goodfellow et al., 2014). We hypothesize that reversing images with high perturbation is much easier because training loss is higher as the difference between perturbed data and original data is much higher, making training converge much faster.

The benefit of reconstructing the image is that the network becomes more robust for various $\epsilon$ outside the training distribution. After observing the recovery accuracy on $\epsilon = 32$, which is way out of the $\epsilon$ range used for training, the image reconstruction method yields up to 75% reclassification accuracy. In contrast, the noise reconstruction method yields up to 30% reclassification accuracy. This difference is due to how image reconstruction differs from noise reconstruction. In the case of the image reconstruction method, the encoder can still extract partial semantic information about the image, which is consequently used to reconstruct it. While noise reconstruction has not been trained to predict the scale of pixel values that will reverse the attack, which in this case is in between [-32,32], and the network was only trained to predict a range between [-16, 16].

## 8. Conclusion

A novel architecture and training procedure is devised where the adversarial images are recovered with enhancing the semantics of the original image making the models classify better than classification capacity for original images. This architecture is capable of recovering images on a large magnitude of attacks. The architecture will be validated on ImageNet on availability of higher compute resources.

## 7. Results

After being trained for the range of 128 $\epsilon$, our generator reconstructed the attacked images. When tested on Inception-v3 and for robustness tested on VGG-13, these images gave staggering results, as seen in Table 2. Both the models were able to successfully classify the reconstructed images where in some cases, the models' results (25 $\epsilon$ - 98.5%) were better than what the models performed on original images (93.1%).

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
