# OpenReview forum: "Robust Recovery of Adversarial Examples"
_ICML.cc/2021/Workshop/AML — ICML 2021 Workshop AML Poster_

### Official Review · Reviewer_rftm · 2021-06-19
**The paper proposed a novel  architecture and training procedure to recover adversarial images and enhance the semantics of the images, but seems not be polished well.**

**Rating:** Accept
**Confidence:** 4

**Review:**

The paper proposed a novel  architecture and training procedure to recover adversarial images and enhance the semantics of the images. The paper also analyzed and compared two image recovery approaches, which are image reconstruction and noise reconstruction.  The experimental results show the proposed method has a good capability of reversing the images and enhancing the semantic meaning of the images.

pros:
1. The paper is easy to follow.
2. The paper provides substantial experiments and analysis to verify the effectiveness of their method.

cons:
1. The paper seems not be polished well. There are some typo and formatting errors. For example, Line 147 Diffusion Models (?), Line 66 E_max, and the citation format does not match the template.
2. Lacks comparative experiments with other adversarial image recovery methods.

---

### Decision · Program_Chairs · 2021-06-21

**Decision:**

Accept (Poster)

**Comment:**

This paper proposed a novel architecture and training procedure to recover adversarial images and enhance the semantics of the images. The paper need further polishing.